# Effectiveness of Influenza Vaccination and Early Antiviral Treatment in Reducing Pneumonia Risk in Severe Influenza Cases

**DOI:** 10.3390/vaccines12020173

**Published:** 2024-02-07

**Authors:** Pere Godoy, Núria Soldevila, Ana Martínez, Sofia Godoy, Mireia Jané, Nuria Torner, Lesly Acosta, Cristina Rius, Àngela Domínguez

**Affiliations:** 1Institut de Recerca Biomédica de Lleida (IRBLleida), Universitat de Lleida, 25006 Lleida, Spain; sofiagodoygarcia@gmail.com; 2CIBER Epidemiología y Salud Pública (CIBERESP), 28029 Madrid, Spain; nsoldevila@ub.edu (N.S.); a.martinez@gencat.cat (A.M.); mireia.jane@gencat.cat (M.J.); nuriatorner@ateneu.ub.edu (N.T.); crius@aspb.cat (C.R.); angela.dominguez@ub.edu (À.D.); 3Departament de Medicina, Universitat de Barcelona, 08036 Barcelona, Spain; 4Agència de Salut Pública de Catalunya, 08005 Barcelona, Spain; 5Institut Català de la Salut, 08007 Lleida, Spain; 6Agència de Salut Pública de Barcelona, 08023 Barcelona, Spain

**Keywords:** influenza, vaccination, pneumonia, antiviral treatment

## Abstract

Introduction: Influenza vaccination may be effective in preventing influenza infection and may reduce the risk of influenza-associated pneumonia. The study aim was to evaluate the effect of influenza vaccination in preventing pneumonia when it failed to prevent influenza hospitalization. Methods: This was a case–control study comparing hospitalized cases of influenza with and without pneumonia in patients aged ≥18 years in 16 hospitals in Catalonia over 10 influenza seasons (2010–11 to 2019–20). Data on sociodemographic, virological characteristics, comorbidities, vaccination history, and antiviral treatment were collected and analysed. The crude odds ratio (OR) and adjusted OR (aOR) with the corresponding 95% confidence interval (CI) values were calculated. Results: In total, 5080 patients hospitalized for severe influenza were included, 63.5% (3224/5080) of whom had pneumonia—mostly men (56.8%; 1830/3224) and mostly in the ≥75 age group (39.3%; 1267/3224)—and of whom 14.0% died (451/3224). Virus A and virus B accounted for 78.1% (2518/3224) and 21.9% (705/3224) of influenza types, respectively. Starting antiviral treatment ≤48 h after symptom onset (aOR = 0.69; 95%CI: 0.53–0.90) and a history of seasonal influenza vaccination (aOR = 0.85; 95%CI: 0.72–0.98) were protective factors in developing pneumonia. Conclusions: Adherence to seasonal influenza vaccination and starting antiviral treatment within 48 h of symptom onset can reduce pneumonia risk in severe influenza cases.

## 1. Introduction

Every year, around 5–20% of the population are infected by the influenza virus, resulting in about 3–5 million cases of severe illness and 300,000–500,000 deaths [1]. The World Health Organization (WHO) has underlined the need to monitor the clinical severity of cases and to collect data on comorbidities to identify factors that contribute to greater severity and mortality [2].

One of the main complications of influenza is community-acquired pneumonia, resulting from direct influenza virus infection of the lung parenchyma or from secondary bacterial infection [3,4,5]. Influenza causes ciliary dysfunction and airway obstruction secondary to increased mucus production and oedema, which may contribute to subsequent bacterial colonization [6,7]. Influenza-associated reduced neutrophil granulocyte production and impaired alveolar macrophage function [6,8] may also contribute to the severity of bacterial pneumonia.

Several studies indicate that starting antiviral treatment early after the onset of symptoms can reduce the risk of hospitalization and death and, conversely, that effectiveness in preventing severity may be reduced when antiviral treatment is not started until 48 h after symptom onset [9,10,11,12].

Vaccination also reduces the risk of severe influenza, and is especially recommended for individuals at greater risk of presenting complications, such as patients with comorbidities and the elderly [13,14]. Annual vaccination effectiveness in preventing influenza infection can be as high as 80% if the vaccine matches the circulating strain during epidemic activity [13,14,15,16] and is associated with more effective treatment and better survival of patients with pneumonia [3,6].

We hypothesize that a history of previous influenza vaccination reduces the risk of pneumonia even if infection is not prevented. Our study aimed to evaluate the effectiveness of the seasonal influenza vaccine in preventing pneumonia in severe hospitalized laboratory-confirmed influenza cases registered by the influenza surveillance system of Catalonia (PIDIRAC) over 10 influenza seasons.

## 2. Methods

We conducted a case–control study of severe hospitalized laboratory-confirmed influenza (SHLCI) cases to compare the characteristics and risk factors associated with SHLCI cases with and without pneumonia.

The monitoring of patients hospitalized for severe laboratory-confirmed influenza began in 2010 in Catalonia, a region in northeastern Spain with 7.5 million inhabitants. The influenza surveillance system of Catalonia (PIDIRAC) included 16 hospitals with a catchment area of 5,610,858 inhabitants (73.8% of the total population of Catalonia), which reported SHLCI cases admitted during each influenza epidemic season (from October to May) [16,17].

The study population consisted of individuals aged ≥18 years hospitalized with severe laboratory-confirmed influenza virus infection in one of the sixteen surveillance hospitals over 10 influenza seasons (2010–2011 to 2019–2020).

We included patients hospitalized for ≥24 h with severe ILI in one of the 16 sentinel hospitals. ILI was defined as a combination of (i) sudden onset of symptoms, (ii) ≥1 of the following symptoms: fever (≥38 °C), headache, myalgia, malaise, and (iii) ≥1 of the following respiratory symptoms: cough, sore throat, dyspnea. Patients were recruited at the participating hospitals by a sentinel physician who evaluated all patients with severe ILI. We defined severe laboratory-confirmed influenza as cases infected with influenza virus that require hospital admission due to their severity (pneumonia, acute respiratory distress syndrome, septic shock, multi-organ failure, or any other serious condition, including admission to the intensive care unit (ICU) or death) [16]. We obtained a nasopharyngeal or pharyngeal swab: bronchoalveolar lavage fluid (BAL) or, for patients admitted to the ICU, tracheal aspirate (TA). Influenza virus infection was detected by reverse transcriptase polymerase chain reaction (RT-PCR).

Cases with pneumonia were defined as patients with severe influenza, aged ≥18 years, with pulmonary infiltrate evidenced by chest X-ray, a history of fever, and at least one sign or symptom of lower respiratory tract infection (cough, sputum, or focal signs of pneumonia on auscultation). Controls were patients hospitalized during the same influenza season as cases presenting with severe laboratory-confirmed influenza but without pneumonia.

Information on cases and controls was collected by epidemiological surveillance staff from interviews and a structured questionnaire (as used by the Catalan Epidemiological Surveillance Network). Medical records were reviewed to collect sociodemographic data and clinical data on obesity (body mass index (BMI) > 40); pregnancy; chronic conditions, including chronic obstructive pulmonary disease (COPD), diabetes, chronic kidney disease (CKD), immunodeficiency (human immunodeficiency virus (HIV) and other infections), chronic cardiovascular disease (CVD), and chronic liver disease (CLD); diagnosis; symptom onset; hospital admission and discharge dates; complications (primary or secondary pneumonia with and without bacterial co-infection, acute respiratory distress syndrome (ARDS), and multiple organ failure); antiviral treatment; influenza vaccination (defined as vaccination at least 14 days before the onset of symptoms); ICU admission; and death.

### 2.1. Laboratory Data

Patients’ clinical samples were analysed by real-time RT-PCR for influenza A and B viruses after manual nucleic acid extraction in the surveillance hospitals’ laboratories.

Amplification was performed in an ABI 7500 thermocycler. Samples with no typed influenza viruses were sent to the Catalan Influenza Reference Centre to determine the type, where adequate positive samples were typed for viruses known to be circulating at the time, namely type A (subtypes H1N1pdm09 and H3N2) and type B. Molecular subtyping determined the HA subtype for influenza A. Two specific one-step multiplex real-time PCR techniques using Stratagene Mx3000P QPCR Systems (Agilent Technologies, Santa Clara, CA, USA) were carried out to type A. Samples for which typing failed due to a low viral load were classified as unidentifiable.

### 2.2. Statistical Analysis

Cases of severe influenza with pneumonia were compared with all other cases of severe influenza considering the following variables: sex, age group (18–64, 65–74, and ≥75 years), influenza virus type, comorbidities (COPD, diabetes, obesity, CKD, immunodeficiency, chronic CVD, and CLD), complications, and antiviral treatment. Categorical variables were compared using the chi-square test and Fisher’s exact test, and continuous variables were compared using a *t*-test.

A bivariate analysis explored the relationships between the dependent variable (severe influenza with/without pneumonia) and the independent variables, for which the crude odds ratio (OR) and corresponding 95% confidence interval (CI) values were calculated. For multivariate logistic regression analysis, based on backward selection of variables according to a cut-off point of *p* < 0.2, the adjusted odds ratio (aOR) and the corresponding 95% CI values were calculated.

Analyses were performed using the SPSS v.24 statistical package and R v3.5.0 statistical software (http://cran.r-project.org, accessed on 1 July 2023).

### 2.3. Ethics Statement

All data used in the analysis were collected during routine public health surveillance activities as part of the legislated mandate of the Health Department of Catalonia, the competent authority for the surveillance of communicable diseases, which is officially authorized to receive, treat, and temporarily store personal data on cases of infectious disease. Therefore, data were exempt from institutional board review and did not require informed consent. All data were completely anonymized before analysis [18].

## 3. Results

Of a total of 5080 patients hospitalized for severe laboratory-confirmed influenza included in the study, 63.5% (3224/5080) had pneumonia and 13.5% died (686/5080). Women accounted for 44.4% (2254/5080) of patients included, and 27.7% (1391/5080) were aged 45–64 years (Table 1).

Most patients (79.3% (4028/5080)) were infected by type A influenza virus (H1N1 = 1230, H3N2 = 1000, and unsubtyped = 1798), and the majority (77.7%) had one or more comorbidities (3948/5080), especially CVD (33.8% (1709/5080)), diabetes (26.6% (1339/5080)), and COPD (26.1% (1324/5080)). Only 32.8% (1632/5080) had received the influenza vaccine, and 58.5% (2856/5080) had started antiviral treatment 48 h after symptom onset (Table 1).

Compared with influenza patients without pneumonia, the group with pneumonia had fewer women (43.2% versus 46.3%), individuals aged 65–74 years (19.9% versus 22.0%) and individuals aged ≥75 years (39.3% versus 47.1%) and had lower rates of COPD (22.5% versus 31.9%), obesity (7.7% versus 11.4%), CKD (15.9% versus 18.6%), and CVD (32.7% versus 35.7%) (Table 2).

However, in the patients with pneumonia, the severity profile was greater, as reflected in the greater risk of ICU admission (25.8% versus 18.8%) and of mortality (14.0% versus 12.7%). Additionally, a higher percentage had started antiviral treatment 48 h after symptom onset (63.0% versus 50.5%), and a smaller percentage had been vaccinated against influenza (30.3% versus 37.2%) (Table 3). The profile of severity and the effectiveness of the vaccine and antivirals was maintained in the analysis stratified by age groups, although, for some characteristics, the differences were not statistically significant due to the lower number of patients in these groups (Appendix A).

In the regression model, the protective factors in developing pneumonia were sex (aOR = 0.82; 95%CI: 0.72–0.94), starting antiviral treatment ≤48 h after symptom onset (aOR = 0.69; 95%CI: 0.53–0.90), and a history of seasonal influenza vaccination (aOR = 0.85; 95%CI: 0.72–0.98) (Table 4).

## 4. Discussion

A high percentage of severe hospitalized laboratory-confirmed influenza cases had pneumonia (63.5%), required ICU admission (23.2%), and resulted in death (13.5%). Despite serious risk factors for influenza, a high percentage of those individuals were only started on antiviral treatment 48 h after symptom onset. Furthermore, although most also had comorbidities as well as risk factors, influenza vaccination coverage was only 32.8%, corroborating findings reported by other studies [14,19].

Starting antiviral treatment in the first 48 h can mitigate the pneumonia risk, while delaying treatment to after 48 h may be a risk factor [20,21]. Antiviral treatment can reduce pneumonia risk and the risk of hospital admission by reducing lesions, and thus inflammation and pulmonary oedema, and by improving immune system response at the pulmonary level. In a meta-analysis that included 3376 patients, Venkatesan et al. [22] estimated 76% effectiveness for neuraminidase inhibitor (NAI) treatment in reducing hospital admission risk and also reported that effectiveness was even greater in patients treated in the first 48 h. In another meta-analysis that included 78 studies, Mothuri et al. [9] reported a reduced mortality risk for both antiviral treatment administered within 2 days of symptom onset compared with delayed treatment (aOR = 0.48; *p* < 0.001) and for early treatment versus no treatment (aOR = 0.50; *p* < 0.001). Dominguez et al. [23], in a study of 1727 hospitalized patients in Catalonia, reported a reduced mortality risk (aOR = 0.37; *p* < 0.001) for patients receiving NAI treatment in the first 48 h after clinical symptom onset. Similar results have been observed in a surveillance data survey conducted in 11 European countries [24]. Nonetheless, despite the evidence available on the effectiveness of antivirals in preventing severity, several barriers to their systematic use exist and clinical practice is quite variable [13,25].

Controlling for the effect of antiviral treatment, in our study, a history of vaccination was 15% effective in reducing the pneumonia risk in severe cases of influenza, a finding similar to that observed in other studies [20,21,22,23,24,25,26,27]. While this result may, on the face of it, seem to be less than spectacular, it needs to be considered in a context of pneumonia representing a greater risk of ICU admission and death for our patients, despite their relatively younger age and despite the greater severity profile for the control group due to higher rates of obesity, COPD, and ARDS.

The study has some limitations. Some patients may have been vaccinated in private centres and so their vaccination status was not in their medical records, although this is unlikely given that the Catalan vaccination programme is free and universal. Since symptom onset is difficult to determine in some patients, an inaccurate record may have delayed antiviral treatment, leading to an underestimation of pneumonia risk, as has been reported elsewhere [24]. While more influenza diagnostic tests may have been ordered for unvaccinated patients, this seems unlikely since healthcare providers were unaware of vaccination histories at the time of ordering the tests. The effectiveness of the influenza vaccine may be underestimated due to the lack of matching among the virus strains in circulation and the vaccine strains in some seasons. Pneumonia in the elderly may be underestimated because it often presents with few symptoms and age may have a confounding effect, but the stratified analysis by age groups yields similar results (Appendix A). The existence of residual confounders cannot be ruled out, although this is unlikely as most of the confounders described in the literature were assessed in the multivariate logistic regression model. As for strengths, the study was performed over 10 epidemic influenza seasons and data were collected by a stable and robust epidemiological surveillance service covering over 60% of the Catalan population.

## 5. Conclusions

Influenza vaccination and starting antiviral treatment within 48 h of symptom onset can reduce pneumonia risk in severe influenza cases.

## Figures and Tables

**Table 1 vaccines-12-00173-t001:** Sociodemographic characteristics, virological data, and comorbidities of patients with severe influenza during the 2010/2011–2019/2020 flu seasons in Catalonia (Spain).

	Severe Hospitalized Laboratory-Confirmed Influenza Patients (SHLCI) N = 5080 (%)
Influenza vaccination (4977)	
Yes	1632 (32.8)
No	3345 (67.2)
Sex	
Female	2254 (44.4)
Male	2826 (55.6)
Age, years	
18–44	495 (9.7)
45–64	1391 (27.4)
65–74	1052 (20.7)
≥75	2142 (42.2)
ICU admission	
Yes	1179 (23.2)
No	3901 (76.8)
Death	
Yes	686 (13.5)
No	4394 (86.5)
≥1 comorbidities	
Yes	3948 (77.7)
No	1132 (22.3)
COPD (5072)	
Yes	1324 (26.1)
No	3748 (73.9)
Obesity (5007)	
Yes	452 (9.0)
No	4555 (91.0)
Diabetes (5042)	
Yes	1339 (26.6)
No	3703 (73.4)
CKD (5066)	
Yes	855 (16.9)
No	4211 (83.1)
Immunodeficiency (5056)	
Yes	913 (18.1)
No	4143 (81.9)
CVD (5060)	
Yes	1709 (33.8)
No	3351 (66.2)
CLD (5047)	
Yes	294 (5.8)
No	4753 (94.2)
Antiviral treatment (5069)	
Yes	4735 (93.4)
No	334 (6.6)
Antiviral treatment (4883)	
≤48 h before symptom onset	1693 (34.7)
>48 h after symptom onset	2856 (58.5)
No	334 (6.8)
Hospital stay, days (5075)	
0–14	3653 (72.0)
>14	1422 (28.0)
ARDS (4884)	
Yes	2070 (42.4)
No	2814 (57.6)
Multiple organ failure (4952)	
Yes	423 (8.5)
No	4529 (91.5)
Influenza virus type	
A	4028 (79.3)
B	1048 (20.6)
C	4 (0.1)
Influenza season	
2010–11	168 (3.3)
2011–12	122 (2.4)
2012–13	118 (2.3)
2013–14	343 (6.7)
2014–15	423 (8.3)
2015–16	546 (10.7)
2016–17	579 (11.4)
2017–18	1241 (24.4)
2018–19	1046 (20.6)
2019–20	494 (9.7)

Abbreviations: ARDS: acute respiratory distress syndrome. CKD: chronic kidney disease. CLD: chronic liver disease. COPD: chronic obstructive pulmonary disease. CVD: cardiovascular disease. ICU: intensive care unit.

**Table 2 vaccines-12-00173-t002:** Comparison of severe influenza patients with and without pneumonia according to sociodemographic characteristics, virological data, and comorbidities during the 2010/2011–2019/2020 influenza seasons in Catalonia (Spain).

	Patients with Pneumonia (N = 3224)	Patients without Pneumonia (N = 1856)	OR (95%CI)	*p* Value
Sex				
Female	1394 (43.2%)	860 (46.3%)	0.88 (0.79–0.99)	0.03
Male	1830 (56.8%)	996 (53.7%)	Ref.	
Age				
18–44	367 (11.4%)	128 (6.9%)	1.34 (1.07–1.69)	0.01
45–64	947 (29.4%)	444 (23.9%)	Ref.	
65–74	643 (19.9%)	409 (22.0%)	0.74 (0.62–0.87)	<0.01
≥75	1267 (39.3%)	875 (47.1%)	0.68 (0.59–0.78)	<0.01
≥1 comorbidities				
Yes	2406 (74.6%)	1542 (83.1%)	0.60 (0.52–0.69)	<0.01
No	818 (25.4%)	314 (16.9%)	Ref.	
COPD				
Yes	725 (22.5%)	589 (31.9%)	0.62 (0.55–0.71)	<0.01
No	2491 (77.5%)	1257 (68.1%)	Ref.	
Obesity				
Yes	243 (7.7%)	209 (11.4%)	0.65 (0.53–0.79)	<0.01
No	2927 (92.3%)	1628 (88.6%)	Ref.	
Diabetes				
Yes	824 (25.7%)	515 (28.1%)	0.89 (0.78–1.01)	0.07
No	2382 (74.3%)	1321 (71.9%)	Ref.	
CKD				
Yes	510 (15.9%)	345 (18.6%)	0.82 (0.71–0.96)	0.01
No	2706 (84.1%)	1505 (81.4%)	Ref.	
Immunodeficiency				
Yes	603 (18.8%)	305 (16.6%)	1.16 (1.00–1.35)	0.05
No	2608 (81.2%)	1535 (83.4%)	Ref.	
CVD				
Yes	1051 (32.7%)	658 (35.7%)	0.88 (0.78–0.99)	0.03
No	2165 (67.3%)	1186 (64.3%)	Ref.	
CLD				
Yes	197 (6.1%)	97 (5.3%)	1.17 (0.91–1.51)	0.21
No	3012 (93.9%)	1741 (94.7%)	Ref.	
Hospital stay, days				
0–14	2324 (72.2%)	1329 (71.6%)	1.03 (0.90–1.17)	0.69
>14	896 (27.8%)	526 (28.4%)	Ref.	
ARDS				
Yes	1029 (32.7%)	1141 (62.0%)	0.30 (0.26–0.34)	<0.01
No	2115 (67.3%)	699 (38.0%)	Ref.	
Multiorgan failure				
Yes	287 (9.2%)	145 (7.9%)	1.18 (0.96–1.45)	0.12
No	2839 (90.8%)	1690 (92.1%)	Ref.	
Viral type				
A	2518 (78.1%)	1510 (81.4%)	Ref.	
B	705 (21.9%)	343 (18.5%)	1.23 (1.07–1.42)	0.004
C	1 (0.0%)	3 (0.2%)	0.20 (0.02–1.92)	0.16
Flu season				
2010–11	127 (3.9%)	41 (2.2%)	3.39 (2.29–5.02)	<0.01
2011–12	94 (2.9%)	28 (1.5%)	3.67 (2.32–5.80)	<0.01
2012–13	87 (2.7%)	31 (1.7%)	3.07 (1.96–4.79)	<0.01
2013–14	222 (6.9%)	121 (6.5%)	2.01 (1.51–2.66)	<0.01
2014–15	330 (10.2%)	93 (5.0%)	3.88 (2.90–5.18)	<0.01
2015–16	432 (13.4%)	114 (6.1%)	4.14 (3.16–5.44)	<0.01
2016–17	356 (11.0%)	223 (12.0%)	1.75 (1.37–2.23)	<0.01
2017–18	785 (24.3%)	456 (24.6%)	1.88 (1.52–2.32)	<0.01
2018–19	555 (17.2%)	491 (26.5%)	1.24 (0.98–1.53)	0.05
2019–20	236 (7.3%)	258 (13.9%)	Ref.	

Abbreviations: ARDS: acute respiratory distress syndrome. CI: confidence interval. CKD: chronic kidney disease. CLD: chronic liver disease. COPD: chronic obstructive pulmonary disease. CVD: cardiovascular disease. OR: odds ratio.

**Table 3 vaccines-12-00173-t003:** Comparison of severe influenza patients with and without pneumonia according to antiviral treatment, influenza vaccination, intensive care unit admission, and death, during the 2010/2011–2019/2020 influenza seasons in Catalonia (Spain).

	Patients with Pneumonia (N = 3224)	Patients without Pneumonia (N = 1856)	OR (95%CI)	*p* Value
Influenza vaccination				
Yes	960 (30.3%)	672 (37.2%)	0.73 (0.65–0.83)	<0.01
No	2212 (69.7%)	1133 (62.8%)	Ref.	
ICU admission				
Yes	831 (25.8%)	348 (18.8%)	1.50 (1.31–1.73)	<0.01
No	2393 (74.2%)	1508 (81.3%)	Ref.	
Death				
Yes	451 (14.0%)	235 (12.7%)	1.12 (0.95–1.33)	0.18
No	2773 (86.0%)	1621 (87.3%)	Ref.	
Antiviral treatment				
Yes	2994 (93.1%)	1741 (94.0%)	0.86 (0.68–1.08)	0.20
No	223 (6.9%)	111 (6.0%)	Ref.	
Antiviral treatment				
≤48 h before symptom onset	927 (29.8%)	766 (43.2%)	0.60 (0.47–0.77)	<0.01
>48 h after symptom onset	1961 (63.0%)	895 (50.5%)	1.09 (0.86–1.39)	0.48
No	223 (7.2%)	111 (6.3%)	Ref.	

Abbreviations: CI: confidence interval. ICU: intensive care unit. OR: odds ratio.

**Table 4 vaccines-12-00173-t004:** Logistic multivariate regression model of pneumonia risk factors for the 2010/2011–2019/2020 influenza seasons in Catalonia (Spain).

	aOR (95% CI)	*p* Value
Influenza vaccination		
Yes	0.85 (0.72–0.98)	0.03
No	Ref.	
Sex		
Female	0.82 (0.72–0.94)	0.01
Male	Ref.	
Age		
18–44	1.20 (0.93–1.55)	0.16
45–64	Ref.	
65–74	0.82 (0.68–0.99)	0.04
≥75	0.74 (0.62–0.87)	<0.01
Antiviral treatment		
≤48 h before symptom onset	0.69 (0.53–0.90)	<0.01
>48 h after symptom onset	1.16 (0.90–1.50)	0.26
No	Ref.	

Abbreviations: aOR: adjusted odds ratio (adjusted for all variables in the table and chronic obstructive pulmonary disease, obesity, immunodeficiency, antiviral treatment, and flu season). CI: confidence interval.

## Data Availability

The dataset is available from the corresponding author upon reasonable request.

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
