# Peer review of "Effectiveness of Influenza Vaccination and Early Antiviral Treatment in Reducing Pneumonia Risk in Severe Influenza Cases"

_vaccines, 2024, doi:10.3390/vaccines12020173_

Round 1

Reviewer 1 Report

Comments and Suggestions for Authors Abstract.

1.       Line 34-36. Included were 5080 patients hospitalized for severe influenza, 63.5% (3224/5080) of whom had pneumonia mostly women (43.2%; 1394/3224) and most in the 18-64 age group (40.8%; 1314/3224) – and of whom 14.0% died (451/3224).

Comment 1: mostly women (43.2%) is not correct in my opinion.

2.       Line 37-38. Virus A and virus B accounted for 78.1% (2518/3224) and 21.9% (705/3224) of influenza subtypes, respectively.

Comment 2: The sentence constructed incorrectly. Only the types of influenza virus are indicated, not subtypes.

3.       Line 38-40.   “ Variables associated with pneumonia were the female sex

(aOR=0.83; 95%CI: 0.73-0.94), COPD (aOR=0.61; 95%CI: 0.53-0.71), obesity (aOR=0.63; 95%CI: 0.50-

0.77), and starting antiviral treatment 48 hours after symptom onset (aOR=1.68; 95%CI: 1.47-1.92)”.

Comment 3: In my opinion, it will be better to indicate specifically which variables turned out to be a risk factor for the pneumonia.

 Introduction.

4.       Line 66-68. Annual vaccination effectiveness in preventing influenza infection can be as high as 80% if the vaccine matches the circulating strain of the circulating virus during epidemic activity.

Comment 4. The repeat needs to be removed (highlighted in yellow).

Methods.

5.       Lines 99-101 and lines 91-92 as well as  lines 110-111 are very similar.

Comment 5. Repetitions must be removed.

6.       Line 123. Virological data were collected on laboratory.

Comment 6. The phrase does not contain specific information. The name of the laboratory and organization should be clarified or it should be deleted.

7.       Lines 127-131. Samples with unsubtyped influenza viruses were sent to the Catalan Influenza Reference Centre to determine the subtype, where positive samples were molecularly subtyped for viruses known to be circulating at the time, namely type A (subtypes H1N1pdm09 and H3N2) and type B. Molecular subtyping determined the H subtype for influenza A and the lineage for influenza B.

Comment 7. It is not clear from the text whether all clinical samples were investigated to influenza A virus subtyping and influenza B virus sublineage determination? What specific kits were used for that, the manufacturer of the kits? Have viruses been sequenced if their subtype could not be determined by RT-PCR? H subtype - Do you mean the hemagglutinin (HA) subtype?

Comment 8. Ethics committee approval for research must be indicated in the Methods.

Results.

8.       Lines 160 – 164 should be placed above table 1.

Comment 9. It would be interesting to clarify which influenza A virus subtypes and influenza B virus lineages circulated in each season. How did this affect the frequency of recorded pneumonia? Have there been similarities (or differences) between the circulating in Catalonia viruses and the vaccine strains?

Conclusions.

9.       Lines 238-242 and lines 42-44.

Comment 10. In my opinion, the Conclusions in the Abstract more accurately reflect the content of the manuscript.

Comments on the Quality of English Language

Minor editing of English language required

Author Response

Answer to Editor and Reviewers

Open Review -1

Comments and Suggestions for Authors

Answer. Thank you for reviewing the article and raising these important questions. We have made further efforts to improve the manuscript.

Abstract.

  1. Line 34-36. Included were 5080 patients hospitalized for severe influenza, 63.5% (3224/5080) of whom had pneumonia mostly women (43.2%; 1394/3224) and most in the 18-64 age group (40.8%; 1314/3224) – and of whom 14.0% died (451/3224).

Comment 1: mostly women (43.2%) is not correct in my opinion.

Answer. According to the reviewer, we have changed the sentence: “mostly men (56.8%; 1830/3224) and most in the 18-64 age group (40.8%; 1314/3224) – and of whom 14.0% died (451/3224).

  1. Line 37-38. Virus A and virus B accounted for 78.1% (2518/3224) and 21.9% (705/3224) of influenza subtypes, respectively.

Comment 2: The sentence constructed incorrectly. Only the types of influenza virus are indicated, not subtypes.

Answer. According to the reviewer, we have changed subtypes by types: “Virus A and virus B accounted for 78.1% (2518/3224) and 21.9% (705/3224) of influenza types, respectively

  1. Line 38-40. “ Variables associated with pneumonia were the female sex

(aOR=0.83; 95%CI: 0.73-0.94), COPD (aOR=0.61; 95%CI: 0.53-0.71), obesity (aOR=0.63; 95%CI: 0.50-0.77), and starting antiviral treatment 48 hours after symptom onset (aOR=1.68; 95%CI: 1.47-1.92)”.

Comment 3: In my opinion, it will be better to indicate specifically which variables turned out to be a risk factor for the pneumonia.

Answer. According to the reviewer, we have added: “Starting antiviral treatment 48 hours after symptom onset was a risk factor to pneumonia (aOR=1.68; 95%CI: 1.47-1.92) and a history of previous seasonal influenza vaccination was a protective factor in developing pneumonia (aOR=0.85; 95%CI: 0.73-0.98).”

 Introduction.

  1. Line 66-68. Annual vaccination effectiveness in preventing influenza infection can be as high as 80% if the vaccine matches the circulating strain of the circulating virus during epidemic activity.

Comment 4. The repeat needs to be removed (highlighted in yellow).

Answer. According to the reviewer, we have removed the repetition

Methods.

  1. Lines 99-101 and lines 91-92 as well as lines 110-111 are very similar.

Comment 5. Repetitions must be removed.

Answer. According to the reviewer, we have removed the repetition

  1. Line 123. Virological data were collected on laboratory.

Comment 6. The phrase does not contain specific information. The name of the laboratory and organization should be clarified or it should be deleted.

Answer. According to the reviewer, the we have deleted

  1. Lines 127-131. Samples with unsubtyped influenza viruses were sent to the Catalan Influenza Reference Centre to determine the subtype, where positive samples were molecularly subtyped for viruses known to be circulating at the time, namely type A (subtypes H1N1pdm09 and H3N2) and type B. Molecular subtyping determined the H subtype for influenza A and the lineage for influenza B.

Comment 7. It is not clear from the text whether all clinical samples were investigated to influenza A virus subtyping and influenza B virus sublineage determination? What specific kits were used for that, the manufacturer of the kits? Have viruses been sequenced if their subtype could not be determined by RT-PCR? H subtype - Do you mean the hemagglutinin (HA) subtype?

Answer. According to the reviewer, we have reviewed the explanation and now we have written:

“Amplification was performed in an ABI 7500 thermocycler. Samples with no typed influenza viruses were sent to the Catalan Influenza Reference Centre to determine the type, where adequate positive samples were typed for viruses known to be circulating at the time, namely type A (subtypes H1N1pdm09 and H3N2) and type B. Molecular subtyping determined the HA subtype for influenza A. Two specific one-step multiplex re-al-time PCR techniques using Stratagene Mx3000P QPCR Systems (Agilent Technologies, Santa Clara, CA, USA) were carried out to type A. Samples for which typing failed due to a low viral load were classified as unidentifiable.”

Comment 8. Ethics committee approval for research must be indicated in the Methods.

Answer. According to the reviewer, we have added

“2.3. Ethics statement

All data used in the analysis were collected during routine public health surveillance activities, as part of the legislated mandate of the Health Department of Catalonia, the competent authority for the surveillance of communicable diseases, which is officially authorized to receive, treat and temporarily store personal data on cases of infectious disease. Therefore, data were exempt from institutional board review and did not require informed consent. All data were fully anonymized. (18)

Results.

  1. Lines 160 – 164 should be placed above table 1.

Comment 9. It would be interesting to clarify which influenza A virus subtypes and influenza B virus lineages circulated in each season. How did this affect the frequency of recorded pneumonia? Have there been similarities (or differences) between the circulating in Catalonia viruses and the vaccine strains?

Answer. We do not have available information on the influenza A virus subtypes and influenza B virus lineages that circulate each influenza season. As a limitation, we have added, “The effectiveness of the influenza vaccine may be underestimated due to the lack of matching between the virus strain in circulation and the vaccine strain in some seasons.”

Conclusions.

  1. Lines 238-242 and lines 42-44.

Comment 10. In my opinion, the Conclusions in the Abstract more accurately reflect the content of the manuscript.

Answer. According to the reviewer, we have added the same conclusions as the Abstract: “Influenza vaccination and starting antiviral treatment within 48 hours of symptom onset can reduce pneumonia risk in severe influenza cases.”

Reviewer 2 Report

Comments and Suggestions for Authors

The manuscript "Effectiveness of influenza vaccination and early antiviral treatment in reducing pneumonia risk in severe influenza cases" reported a retrospective case-control study to evaluate the effect 28 of influenza vaccination on preventing pneumonia. This study compiled 5080 patient information over a ten-season period. Here are my suggestions.

1. In lines 35-36, the author wrote:"63.5% (3224/5080) of whom had pneumonia – 35 mostly women (43.2%; 1394/3224)". However, according to this data, 43.2% of patients with pneumonia are women, which is less than half of the percentage of men. So this is either a false statement, or the sentence was very confusing.

2. line 38- 40 " Variables associated with pneumonia were the female sex 38 (aOR=0.83; 95%CI: 0.73-0.94), COPD (aOR=0.61; 95%CI: 0.53-0.71), obesity (aOR=0.63; 95%CI: 0.50- 39 0.77), and starting antiviral treatment 48 hours after symptom onset (aOR=1.68; 95%CI: 1.47-1.92).  The authors state variables associated with pneumonia here. However, the OR >1 should be interpreted as positively associated with exposure, and OR<1 should be interpreted as negatively associated with exposure. The statement here is very unclear.  This very confusing data interpretation was also seen in the result part. 

3. Table 2 is very confusing or may be incorrect too. It would be helpful if the author could share the calculation of the ORs. The interpretation of these results is, again, very confusing or may be incorrect. I would suggest that the author review these data carefully. 

4.  references:

a. The format of reference may need to be checked.

b. all the statements from previous knowledge/studies need to reference where the statement came from. for example, Lines 48-49 is a statement with numbers, which should cite the reference the source of data. There are multiple similar issues throughout the paper. please carefully revise this.

This is a large study that could generate great results with correct interpretation. however the current version can not reach the goal yet. Please carefully check each calculation and interpretation of the data to make the study publishable.

Thank you.

Comments on the Quality of English Language

The writing is not concise. The interpretation of results is incorrect or very confusing. I would suggest a major revision for most of the content.

Author Response

Answer to Editor and Reviewers

Open Review-2

Comments and Suggestions for Authors

 The manuscript "Effectiveness of influenza vaccination and early antiviral treatment in reducing pneumonia risk in severe influenza cases" reported a retrospective case-control study to evaluate the effect 28 of influenza vaccination on preventing pneumonia. This study compiled 5080 patient information over a ten-season period. Here are my suggestions.

Answer. Thank you for reviewing the article and raising these important questions. We have made further efforts to improve the manuscript.

  1. In lines 35-36, the author wrote:"63.5% (3224/5080) of whom had pneumonia – 35 mostly women (43.2%; 1394/3224)". However, according to this data, 43.2% of patients with pneumonia are women, which is less than half of the percentage of men. So this is either a false statement, or the sentence was very confusing.

Answer. According to both reviewers, we have changed the sentence: “mostly men (56.8%; 1830/3224) and most in the 18-64 age group (40.8%; 1314/3224) – and of whom 14.0% died (451/3224).

  1. line 38- 40 " Variables associated with pneumonia were the female sex 38 (aOR=0.83; 95%CI: 0.73-0.94), COPD (aOR=0.61; 95%CI: 0.53-0.71), obesity (aOR=0.63; 95%CI: 0.50- 39 0.77), and starting antiviral treatment 48 hours after symptom onset (aOR=1.68; 95%CI: 1.47-1.92). The authors state variables associated with pneumonia here. However, the OR >1 should be interpreted as positively associated with exposure, and OR<1 should be interpreted as negatively associated with exposure. The statement here is very unclear. This very confusing data interpretation was also seen in the result part.

Answer. According to both reviewers, we have added: “Starting antiviral treatment 48 hours after symptom onset was a risk factor to pneumonia (aOR=1.68; 95%CI: 1.47-1.92) and a history of previous seasonal influenza vaccination was a protective factor in developing pneumonia (aOR=0.85; 95%CI: 0.73-0.98).”

  1. Table 2 is very confusing or may be incorrect too. It would be helpful if the author could share the calculation of the ORs. The interpretation of these results is, again, very confusing or may be incorrect. I would suggest that the author review these data carefully.

Answer. We have divided the table 2 into two tables to separate the description of the cases and controls (table 2) from the factors associated with severe flu patients with pneumonia (table 3) and have reviewed the data.

  1. references:
  2. The format of reference may need to be checked.

Answer. According to the reviewer, we have checked the references

  1. all the statements from previous knowledge/studies need to reference where the statement came from. for example, Lines 48-49 is a statement with numbers, which should cite the reference the source of data. There are multiple similar issues throughout the paper. please carefully revise this.

Answer. According to the reviewer the whole text, we have added the references 1, 5, 11, 12 and 18

This is a large study that could generate great results with correct interpretation. however the current version can not reach the goal yet. Please carefully check each calculation and interpretation of the data to make the study publishable.

Answer. According to the reviewers, we have added their suggestions and revised the conclusions. We hope that the reviewers find the article suitable for publication.

Thank you.

Round 2

Reviewer 2 Report

Comments and Suggestions for Authors

Thank you for letting me review the updated version. It is surely improved. Thank you for your hard work. Here are my suggestions.

1. Thank you for correcting line 35 from women to men; however, in line 202, the authors still state, "the variables associated with pneumonia cases were female 202 sex (aOR=0.83; 95%CI: 0.73-0.94)". The OR is <1 so it actually means the females somehow have less chance of acquiring pneumonia. This statement is not correct according to the data.

2. Personally, I think the analysis for age group and antiviral treatment is inappropriate. I would suggest the analysis of age groups use the all-age average as a ref, not one of the age groups, so you can see how each age group, including the 18-64 group, related to the same ref group (all age group). For the antiviral treatment analysis, I would suggest using the non-treatment group as a ref so you can see how the>48h treatment and <48h treatment groups are related to the same ref.

3. line 69-71, I would suggest only keeping the hypothesis as follows: "We hypothesize that the influenza virus favors the development of pneumonia in patients with severe influenza and that a history of previous influenza vaccination reduces the risk of pneumonia even if infection is not prevented" to stay clear and focus on the facts.

4. Dividing tables 2 as tables 2 and 3 may help to emphasize the variable you would like to highlight, but the title of both tables did not represent the information in the table well and did not deliver the main point to the reader. Please revise. 

Thank you very much

Author Response

Answer to Editor and Reviewers

Thank you for letting me review the updated version. It is surely improved. Thank you for your hard work. Here are my suggestions.

Answer. Thank you again for reviewing the new draft of the article and raising these new questions. We have made further efforts to improve the manuscript.

  1. Thank you for correcting line 35 from women to men; however, in line 202, the authors still state, "the variables associated with pneumonia cases were female 202 sex (aOR=0.83; 95%CI: 0.73-0.94)". The OR is <1 so it actually means the females somehow have less chance of acquiring pneumonia. This statement is not correct according to the data.

Answer. According to the reviewer, we have changed the sentence:In the regression model, the variables associated with pneumonia cases were female sex (aOR=0.83; 95%CI: 0.73-0.94) and starting antiviral treatment 48 hours after symptom onset (aOR=1.68; 95%CI: 1.47-1.92). A protective factors in developing pneumonia were sex (aOR=0.82; 95%CI: 0.72-0.94), starting antiviral treatment ≤48 hours after symptom onset (aOR=0.69; 95%CI: 0.53-0.90) and a history of seasonal influenza vaccination (aOR=0.85; 95%CI: 0.72-0.98) (Table 4).”

  1. Personally, I think the analysis for age group and antiviral treatment is inappropriate. I would suggest the analysis of age groups use the all-age average as a ref, not one of the age groups, so you can see how each age group, including the 18-64 group, related to the same ref group (all age group). For the antiviral treatment analysis, I would suggest using the non-treatment group as a ref so you can see how the>48h treatment and <48h treatment groups are related to the same ref.

Answer. According to the reviewer, we have reanalyzed the age and antiviral treatment. We have do another classification of age group (18-44, 45-64, 65-74 and 75) and we have calculated the OR making 45-64 age group as the reference category (table 1, 2 and 4). For the antiviral treatment analysis, we have used the non-treatment group as the reference category (tables 3 and 4) and we have updated the numbers along the text.

  1. line 69-71, I would suggest only keeping the hypothesis as follows: "We hypothesize that the influenza virus favors the development of pneumonia in patients with severe influenza and that a history of previous influenza vaccination reduces the risk of pneumonia even if infection is not prevented" to stay clear and focus on the facts.

Answer. According to the reviewer, we have deleted this sentence and now we have written “We hypothesize that the influenza virus favours the development of pneumonia in patients with severe influenza and that a history of previous influenza vaccination reduces the risk of pneumonia even if infection is not prevented.

  1. Dividing tables 2 as tables 2 and 3 may help to emphasize the variable you would like to highlight, but the title of both tables did not represent the information in the table well and did not deliver the main point to the reader. Please revise.

Answer. According to the reviewer, we have reviewed the title of tables 2 and 3 to make the more informative and now the titles are:

“Table 2. Comparison of severe influenza patients with and without pneumonia according to sociodemographic characteristics, virological data and comorbidities during the 2010/2011-2019/2020 influenza seasons in Catalonia (Spain).”

“Table 3. Comparison of severe influenza patients with and without pneumonia according to antiviral treatment, influenza vaccination, intensive care unit admission and death during the 2010/2011-2019/2020 influenza seasons in Catalonia (Spain).”

Thank you very much